

# The Emissions Model Intercomparison Project (Emissions-MIP): quantifying model sensitivity to emission characteristics

Hamza Ahsan[1], Hailong Wang[2], Jingbo Wu[3], Mingxuan Wu[2], Steven J. Smith[1], Susanne Bauer[3], Harrison Suchyta[1], Dirk Olivié[4], Gunnar Myhre[5], Hitoshi Matsui[6], Huisheng Bian[7], Jean-François Lamarque[8], Ken Carslaw[9], Larry Horowitz[10], Leighton Regayre[9,11], Mian Chin[7], Michael Schulz[4], Ragnhild Bieltvedt Skeie[5], Toshihiko Takemura[12], Vaishali Naik[10]

[1]Joint Global Change Research Institute, Pacific Northwest National Laboratory, College Park, MD, USA
[2]Atmospheric Sciences and Global Change Division, Pacific Northwest National Laboratory, Richland, WA, USA
[3]NASA Goddard Institute for Space Studies, New York, NY, USA
[4]Norwegian Meteorological Institute, Oslo, Norway
[5]CICERO Center for International Climate Research, Oslo, Norway
[6]Graduate School of Environmental Studies, Nagoya University, Nagoya, Japan
[7]NASA Goddard Space Flight Center, Greenbelt, MD, USA
[8]Climate and Global Dynamics Laboratory, National Center for Atmospheric Research, Boulder, CO, USA
[9]Institute for Climate and Atmospheric Science, School of Earth and Environment, University of Leeds, Leeds, UK
[10]NOAA Geophysical Fluid Dynamics Laboratory, Princeton, NJ, USA
[11]Met Office Hadley Centre, Exeter, Fitzroy Road, Exeter, Devon, UK
[12]Research Institute for Applied Mechanics, Kyushu University, Fukuoka, Japan

*Correspondence to*: Hamza Ahsan (hamza.ahsan@pnnl.gov)

**Abstract.** Anthropogenic emissions of aerosols and precursor compounds are known to significantly affect the energy balance of the Earth-atmosphere system, alter the formation of clouds and precipitation, and have substantial impact on human health and the environment. Global models are an essential tool for examining the impacts of these emissions. In this study, we examine the sensitivity of model results to the assumed height of $SO_2$ injection, seasonality of $SO_2$ and BC emissions, and the assumed fraction of $SO_2$ emissions that is injected into the atmosphere as $SO_4$ in 11 climate and chemistry models, including both chemical transport models and the atmospheric component of Earth system models. We find a large variation in atmospheric lifetime across models for $SO_2$, $SO_4$, and BC, with a particularly large relative variation for $SO_2$, which indicates that fundamental aspects of atmospheric sulfur chemistry remain uncertain. Of the perturbations examined in this study, the assumed height of $SO_2$ injection had the largest overall impacts, particularly on global mean net radiative flux (maximum difference of -0.35 W m$^{-2}$), $SO_2$ lifetime over northern hemisphere land (maximum difference of 0.8 days), surface $SO_2$ concentration (up to 59% decrease), and surface sulfate concentration (up to 23% increase). Emitting $SO_2$ at height consistently increased $SO_2$ and $SO_4$ column burdens and shortwave cooling, with varying magnitudes, but had inconsistent effects across models on the sign of the change in implied cloud forcing. The assumed $SO_4$ emission fraction also had a significant impact on net radiative flux and surface sulfate concentration. Because these properties are not standardized across models this is a source of inter-model diversity typically neglected in model intercomparisons. These results imply a need to assure that anthropogenic emission injection height and $SO_4$ emission fraction are accurately and consistently represented in global models.

## 1 Introduction

Anthropogenic emissions of aerosols or their precursors impact atmospheric energy balance, alter the formation of clouds and precipitation, and have substantial impacts on human health and the environment. Global models are an essential tool used to examine the impacts of these emissions. Model results will depend on both the actual input emissions data and the way that data is processed for use, which varies among different modeling systems. Previous work has demonstrated that the assumed injection



height of anthropogenic $SO_2$ emissions has a large impact on modeled surface concentrations in one model (Yang et al. 2019). Here we extend these results in a multi model sensitivity exercise (Emissions-MIP) to explore sensitivity to several aerosol emission-related characteristics across a range of atmospheric models.

Large emission sources, such as anthropogenic point sources and large open fires (Paugam et al., 2016), inject emissions into a heated plume which rises and disperses into the atmosphere. This means that not only are those emissions effectively injected into the atmosphere at some height above the surface, but also the emissions plume may undergo chemical reactions before the atmospheric dispersion. Appropriate distribution of emissions across vertical model layers is necessary to correctly reproduce the atmospheric chemistry in polluted regions (Pozzer et al., 2009).

While injection height for open fires has been a focus of previous studies (Wilkins et al., 2022; Zhu et al., 2018; Val 50 Martin et al., 2018; Paugam et al., 2016), the impact of injection height for anthropogenic emissions in global models has rarely been addressed. Yang et al. (2019), examining the impact of injection height for anthropogenic sulfur ($SO_2$ and $SO_4$), black carbon (BC), and primary organic matter (POM) in the Community Atmosphere Model version 5 (CAM5), found that the effective emission height has a significant impact on the vertical profile and near-surface concentration of $SO_2$ as well as BC and POM. While many regional atmospheric models incorporate plume rise parameterizations, a study on plume rise of $SO_2$ emissions emitted 55 by flare stacks in the Athabasca oil sands found that the commonly used Briggs plume rise algorithm (Briggs, 1982) underpredicted the plume heights of these sources, with up to 52% of the parameterized heights being less than half of the observed height (Akingunola et al., 2018), which ranged from ~500 to ~1,500 m.

Another area of uncertainty in modeling sulfur chemistry is the assumed fraction of the emitted $SO_2$ that is oxidized to $SO_4$ in the atmosphere either at the point of emission or through in-plume processing. Current global- and regional-scale models 60 are generally incapable of accurately resolving aerosol formation within concentrated $SO_2$ sources (Stevens and Pierce, 2013). Therefore, the general approach taken by these models is to assume a fraction of anthropogenic $SO_2$ emissions are emitted into the model grid as sulfate (Makkonen et al., 2009), an assumption that varies between modeling groups. Several studies have investigated the sensitivity of cloud condensation nuclei (CCN) concentrations to changes in the fraction of anthropogenic $SO_2$ assumed to be effectively emitted as sulfate (Luo and Yu, 2011; Wang and Penner, 2009). The consensus from these studies is that 65 particle nucleation rate and size distribution, CCN concentration, and aerosol indirect forcing, are highly sensitive to changes in sulfate fraction and that improved representation of sub-grid scale sulfate formation in global and regional models is required.

Moreover, variations in the temporal and spatial resolution of emissions data can have a significant effect on chemical transport and reaction rates and can potentially impact the climate response in models (Sofiev et al., 2013). One deficiency in the emissions data used in current models, for example, is the inconsistent representation of sub-annual emission rates. A study on 70 Arctic BC concentrations found that in January, the Arctic-mean surface concentrations of BC due to residential combustion emissions were 150% higher when daily emissions were used compared to constant annual emissions (Stohl et al., 2013). Another study used a global chemistry transport model to investigate the sensitivity of temporal variations using the European Monitoring and Evaluation Programme (EMEP) emission inventory and found that the seasonal distribution of emissions had a strong impact on simulated sulfate aerosols, BC and POM (de Meij et al., 2006). For instance, the use of annual average emissions led to an 75 increase in $SO_2$ concentration in June (from 1.57 ppb to 2.26 ppb at one particular location) since residential and commercial heating is less prominent during the summer than in winter.

What is lacking is an examination of how these assumptions impact results across different global models. In this study, therefore, we examine the sensitivity of model results to the assumed height of $SO_2$ injection, seasonality of $SO_2$ and BC, and the assumed fraction of $SO_2$ that is injected into the model as $SO_4$. We expand on previous work by exploring a set of perturbations in 80 11 models, including both chemical transport models and the atmospheric components of Earth system models. The objective is



to quantify the influence of these emission characteristics on model simulations and to better understand the extent to which these characteristics affect results in a similar manner across models. In the following section we outline the models participating in the study and the experimental protocol and provide an overview of the perturbation experiments. Section 3 presents the model simulation results and related analysis. Section 4 presents the key conclusions of the study and discusses the implications of the

results, as well as limitations and potential future work.

## 2 Data and Methods

In this section, we first introduce the 11 global models used in this study (Sect. 2.1). Section 2.2 outlines the experimental protocol and relevant parameters for each of the emission perturbation scenarios. Section 2.3 offers a discussion of why the sensitivities were selected for each perturbation. Finally, Section 2.4 contains a description of the data processing tools and analysis performed.

### 2.1 Models

This study uses output from 11 climate-aerosol and chemical transport models (CTMs) participating in Emissions-MIP. The simulation set-up uses atmosphere-only model runs with prescribed sea surface temperatures (SST) and sea ice concentrations, as well as nudged winds for atmospheric general circulation models (AGCMs) and prescribed meteorology for CTMs. A summary of model characteristics is provided in Table 1.


**Table 1: Models used in this study including relevant model characteristics.**

| Model Abbreviation | Model Version | Nominal Resolution | Vertical Levels | Mid-Latitude Atmos Layer thickness (1st 4 @ ~45°, or all < 400m) over ocean | Reanalysis Atmospheric Data | Ocean Surface Temperature Data | Interactive Aerosol-Meteorology | Endogenous Oxidants | Endogenous DMS Emissions | Key References |
|---|---|---|---|---|---|---|---|---|---|---|
| CESM | CAM5-MAM4 | 1.9° × 2.5° | 30 | 124, 149, 173, 197m | MERRA-2 | HadSST | Yes | No | No | Hurrell et al., 2013; Liu et al., 2016; Yang et al., 2019 |
| E3SM | v1.0 | 1° × 1° | 72 | 25, 54, 72, 77, 82, 87m | MERRA-2 | HadSST | Yes | No | No | Golaz et al., 2019; Rasch et al., 2019; Wang et al., 2020 |
| GISS modelE | E2.1 | 2.5° × 2° | 40 | 170, 190, 220, 240m | MERRA-2 | HadSST | Yes, MATRIX | Yes | Yes | Bauer et al., 2020; Kelley et al., 2020; Bauer et al., 2008 |
| NorESM2 | LM | 1.9° × 2.5° | 32 | 127, 152, 176, 201m | ERA-Interim | HadSST | Yes | Yes | Yes | Seland et al., 2020; Kirkevåg et al., 2018 |
| GFDL-ESM4 | ESM4.1.1 | 1° × 1.25° (100 km cubed sphere) | 49 | 35, 50, 75, 90, 120m | NCEP | PCMDI-AMIP 1.1.2 | Yes | Yes | Yes | Horowitz et al., 2020 |
| CESM2 | WACCM6-MAM4 | 0.9° × 1.25° | 88 | 150, 150, 150, 150m | MERRA-2 | HadSST | Yes | No | No | Emmons et al., 2020; Gettelman et al., 2020 |
| OsloCTM3 | OsloCTM3v1.02 | 2.25° × 2.25° | 60 | 17, 25, 36, 51, 68, 87, 107m | Open IFS ECMWF | Open IFS ECMWF | No | Yes | Yes | Lund et al., 2018; Søvde et al., 2012 |
| GEOS | Icarus-3_3_p2 | 1° × 1° | 72 | 58, 131, 65, 133m | MERRA-2 | MERRA_sst | Yes | No | No | Bian et al., 2017; Colarco et al., 2010; Chin et al., 2000 |
| MIROC-SPRINTARS | MIROC6 | 0.5625° × ~0.5625° | 40 | 21, 49, 71, 92m | ERA-Interim | HadSST | Yes | No | Yes | Takemura et al., 2009; Takemura, 2005 |
| UKESM1 | UKESM1-GC3.1 | 1.25° × 1.875° | 85 | 20, 53, 100, 160, 233, 320m | ERA-Interim | HadSST | Yes | Yes | Yes | Regayre et al., 2022; Mulcahy et al., 2020; Sellar et al., 2019; Williams et al., 2018 |
| CAM-ATRAS | CAM5-ATRAS2 | 1.9° × 2.5° | 30 | 129, 154, 180, 204m | MERRA-2 | HadSST | Yes | Yes | No | Matsui, 2017; Matsui and Mahowald, 2017 |



**2.2 Experiments**

Each modelling group simulated the impact of five perturbations summarized in Table 2. These characteristics are either
inconsistently represented in emission datasets (seasonality) or are inconsistently implemented in individual models (effective
injection height, emitted $SO_4$ fraction). Each experiment uses atmosphere-only model simulations running for a five-year period
from 2000 to 2004 following the year 1999 spin-up as needed by each model. Refer to supplementary file Emissions-MIP
Experimental Protocol - v1b.xlsx for a more detailed breakdown of the model settings for each experiment. The reference case that
is used as the base experiment for comparison consists of the reference state conditions indicated in Table 2.

**Table 2: Reference and perturbation experiments.**

| Emission characteristics | Reference state | Emission perturbation case |
|---|---|---|
| $SO_2$ emission at height | Surface Emissions | (1) All land $SO_2$ emissions emitted between 200 – 400m above land surface (shipping 100 – 300m) |
| %$SO_2$ emitted as $SO_4$ | 2.5% as $SO_4$ | (2) 0%, (3) 7.5% as $SO_4$ |
| $SO_2$ seasonality | CMIP6 (CEDS) seasonality | (4) No $SO_2$ seasonality |
| BC seasonality | CMIP6 (CEDS) seasonality | (5) No BC seasonality |

**2.3 Overview of Perturbation Assumptions**

This section is a review of the set-up for the perturbations examined in the study and discusses the motivation for choosing the
specific sensitivity parameters used in each experiment. The base emissions data for these experiments are anthropogenic emissions
as produced by the Community Emissions Data System (CEDS) for CMIP6 (Hoesly et al., 2018). Anthropogenic emissions as
defined here exclude emissions from open burning of grasslands, forests, and agricultural residues on fields.

**2.3.1 SO₂ emission at height**

Accurate emission data are dependent on spatial resolution and the vertical distribution of the emissions (Pozzer et al., 2009).
However, an underlying cause of uncertainty is the injection height of anthropogenic emissions in global models (Yang et al.,
2019). Most studies that have examined the impact of injection height of anthropogenic emissions used regional models
(Akingunola et al., 2018; Mailler et al., 2013). Pozzer et al. (2009) examined the impact of applying a vertical distribution to
anthropogenic emissions using a global atmospheric chemistry model. Although a strong height dependence was observed for
$NO_x$, CO, NMVOCs, and $O_3$, the impact of vertical distribution on $SO_2$ emissions was not considered in that study. This is a
significant limitation since $SO_2$ is sensitive to vertical distribution to a greater extent than other species (Bieser et al., 2011). Yang
et al. (2019) quantified the effect from injection heights uncertainty of anthropogenic emissions in CAM5, a global aerosol-climate
model. Simulations conducted in that study indicated that the assumed effective emission height (i.e., stack height combined with
plume rise) had a large influence on $SO_2$ near-surface concentrations and vertical profile in CAM5. It was found that the range of
near-surface $SO_2$ concentration over land due to uncertainty in industrial emission injection height was 81% relative to the average
concentration. This result raises the question of whether the sensitivity to injection height is similar across models, and if so, to
what extent.

Any factor that impacts $SO_2$ surface concentrations will also have implications for evaluating models against observations
(at the surface or column burdens retrieved by satellites). Since direct $SO_2$ concentration measurements are mostly available at the



surface, any attempt to validate the sulfur chemistry in the model will be impacted by the injection height assumptions (Johnson et al., 2020). Therefore, systematic assignment of emission data to vertical model layers is important (Pregger and Friedrich, 2009).

Global climate and chemistry models generally rely on assumptions of the height dependency of anthropogenic emissions, such as from the AeroCom (Aerosol Comparisons between Observations and Models) simulation protocol (de Meij et al., 2006; Stier et al., 2005). According to the AeroCom protocol, emissions from industrial facilities and power plants should be injected evenly at a height of 100 to 300 m above the surface, and emissions from international shipping are injected into the lowest model layer (Dentener et al., 2006). No recommendation on assumptions for effective emission injection height was provided as part of CMIP6.

However, the height of plume rise has been measured to exceed these assumed heights, by up to 1 km as was the case for $SO_2$ emissions emitted by flare stacks in the Athabasca oil sands (Akingunola et al., 2018; Gordon et al., 2018). While this is only one example, it indicates both that the effective injection height for anthropogenic sources can be quite large and that there is substantial variability due to changes in meteorology. Ship stacks may be underestimated with respect to their height, as the largest ships (e.g., Panamax) could have a maximum height of 60 m above sea level (Chosson et al., 2008). The plume rise may then extend the

emission height by several hundred meters.

Therefore, for the sensitivity case used here we specify slightly higher effective injection heights compared to those used in the AeroCom study. For the Emissions-MIP emission height perturbation anthropogenic $SO_2$ (and associated $SO_4$) over land was specified to be distributed over 200 – 400 m above the land surface and the shipping sector emissions were distributed over 100 – 300 m above the ocean surface. Emission amounts were assumed to be distributed evenly across the specified altitude range

and proportionally allocated to the relevant model layers.

### 2.3.2 Emitted sulfate fraction

A number of studies have focused on sulfur chemistry within sulfur-rich plumes, as these are a large fraction of anthropogenic aerosols (Wei et al., 2022; Stevens and Pierce, 2013). Global- and regional-scale models are generally unable to accurately resolve aerosol formation within these plumes using grid cells that are tens of kilometers in size or more (Fast et al., 2022; Stevens and

Pierce, 2013). It is typical for these models to assume that a fraction of anthropogenic $SO_2$ emissions is emitted into the model grid as sulfate. For instance, the AeroCom protocol suggests that 2.5% of sulfur should be emitted as sulfate, where most sulfur is emitted as $SO_2$ (Dentener et al., 2006). There have been several sensitivity studies that changed the fraction of emitted $SO_2$ converted to sub-grid sulfate in order to investigate the impact on CCN concentrations (Luo and Yu, 2011; Wang and Penner, 2009). Luo and Yu (2011) found that increasing this fraction from 0 to 5% yielded a change in global boundary layer CCN0.2 (i.e.,

CCN number concentration at 0.2% supersaturation) by 11%. Wang and Penner (2009) demonstrated that even a moderate increase in the $SO_2$ fraction converted to sub-grid sulfate from 0 to 2% resulted in an increase in CCN0.2 by 23% in the boundary layer. Both studies highlighted the importance of accurate parameterizations of sub-grid scale sulfate formation in global aerosol models.

The work cited above focusses on strong emission sources from sulfur-rich plumes. However, these are becoming less commonplace as $SO_2$ emission controls become more stringent. Current emission controls focus on removing solid particulates

and gaseous $SO_2$. Wu et al. (Wu et al., 2020) find that 18% of the sulfur emitted at the stack can be in the form of either filterable or condensable particulates. Further conversion to sulfate occurs in stack plumes (Ding et al., 2021; Luria et al., 2001), which has long been observed to be linear in many cases (Luria et al., 2001) but may be more rapid in wet plumes (Ding et al., 2021). Further, as noted later, a large portion of the emitted sulfur is in the form of $SO_3$.

If we were to assume that 30% of the sulfur from power plants in China is in the form of $SO_3$, then one could have an

aggregate $SO_3$ fraction (for all sectors) over China of up to 8 – 9% (in S mass units). This suggests that, at least in some instances, a much higher fraction of $SO_2$ should be assumed to be emitted as sulfate in global models.



Anthropogenic emission inventories typically specify a total amount of sulfur emissions (as $SO_2$). For the present study, we examined two sensitivity cases for $SO_2$ to $SO_4$ sub-grid conversion, i.e., a "no $SO_4$" case and a "high $SO_4$" case which are specified to have 0% and 7.5% (as %S) anthropogenic $SO_2$ emitted as sulfate, respectively. Emissions of $SO_2$ are reduced
proportionately so as to preserve the total emitted mass of sulfur.

### 2.3.3 Seasonality

Another source of uncertainty in emission data is the temporal distribution, namely, seasonality (i.e., monthly patterns). We note that diurnal and weekly patterns can also influence results; however, these are not evaluated in this work. Aerosol formation and transport (Stohl et al., 2013), as well as chemical reaction rate (Sofiev et al., 2013; Pregger and Friedrich, 2009), are dependent on
the season. Therefore, aerosol and precursor species can have a longer or shorter lifetime depending on the emission seasonality in the model. The emissions data used for CMIP6 (Hoesly et al., 2018) incorporated estimates of seasonality for all sectors and emissions, while the data for prior CMIP phases had partial or no seasonality information. It will be useful to evaluate this aspect of the data, to inform our understanding of the role of aerosols in earlier CMIP experiments.

Aside from openly occurring forest or grass fires which are typically a large source of BC emissions during the summer,
combustion of biomass such as residential wood for heating homes during the winter is a significant source of BC seasonality (Healy et al., 2017). The other major driver of seasonality in aerosol or precursor emissions is space cooling (e.g., air conditioning), which results in some seasonality in electric power production (Sofiev et al., 2017). There is significant seasonality in emissions associated with biological processes, in particular ammonia (Wang et al., 2021), although we did not evaluate this here because that requires models that have sufficiently detailed chemistry.

The two sensitivity scenarios that were considered were identical monthly (averaged annually) emission fluxes for all anthropogenic $SO_2$ emissions (including associated $SO_4$) and anthropogenic BC emissions as compared to the seasonality used in CMIP6, which is used in the reference case (Table 2).

### 2.4 Data processing

Much of the basic data processing in this study was performed with the Earth System Model Evaluation Tool, ESMValTool v2.1.1
(Andela et al., 2020), an open-source diagnostic tool available for the evaluation of Earth system models (Eyring et al., 2020). Simulation results were made available by the participating model groups as netCDF files and, where necessary, processed to conform to the CMIP format (i.e., the data have been "cmorized") for use with ESMValTool. Minor issues in the netCDF files (e.g., missing metadata) were corrected using tools such as netCDF Operator (NCO) or Climate Data Operators (CDO). The datasets from the E3SM, CESM and CESM2 models were cmorized using e3sm_to_cmip, an open-source tool that converts E3SM
(and CESM) model output variables to the CMIP format (Baldwin et al., 2021).

The ESMValTool workflow is controlled by a "recipe" file that defines the datasets, preprocessor options, and diagnostics. All model results were interpolated to 1°×1° grids and the annual mean taken either over the globe or masked to specific region or ocean basin (Figure S1). This functionality was used to compare the impact of emission characteristics in different regions. Each model simulation provided variables for gas and aerosol concentrations and deposition rates as well as radiative fluxes at the
surface and top of the atmosphere. Table 3 provides a list of the variables used in the analysis.

**Table 3: Diagnostics extracted or calculated from model simulations.**

| Diagnostic | CMOR Variable/Formula | Units |
| --- | --- | --- |



| mass mixing ratio of $SO_2$, $SO_4$, BC | so2, mmrso4, mmrbc | kg kg$^{-1}$ |
|---|---|---|
| column burden of $SO_2$, $SO_4$, BC | loadso2, loadso4, loadbc | kg m$^{-2}$ |
| dry deposition rate of $SO_2$, $SO_4$, BC | dryso2, dryso4, drybc | kg m$^{-2}$ s$^{-1}$ |
| wet deposition rate of $SO_2$, $SO_4$, BC | wetso2, wetso4, wetbc | kg m$^{-2}$ s$^{-1}$ |
| total emission rate of $SO_2$ | emiso2 | kg m$^{-2}$ s$^{-1}$ |
| $SO_2$ lifetime | loadso2/emiso2 | days |
| $SO_4$ lifetime | loadso4/(dryso4 + wetso4) | days |
| BC lifetime | loadbc/(drybc + wetbc) | days |
| TOA incident shortwave radiative flux | rsdt | W m$^{-2}$ |
| TOA longwave radiative flux | −rlut | W m$^{-2}$ |
| TOA shortwave radiative flux | rsdt − rsut | W m$^{-2}$ |
| TOA clear-sky longwave radiative flux | −rlutcs | W m$^{-2}$ |
| TOA clear-sky shortwave radiative flux | −rsutcs | W m$^{-2}$ |
| net radiative flux | rsdt − rlut − rsut | W m$^{-2}$ |
| implied cloud radiative flux | rsdt − rlut − rsut + rlutcs + rsutcs | W m$^{-2}$ |
| boundary layer depth | bldep | m |

## 3 Results

In this section we assess the extent to which the perturbation results differ from the reference scenario as well as the spread of
response in models for each experiment. Section 3.1 focuses on the lifetime diagnostics, namely sulfur and BC lifetimes. Section
3.2 provides an overview of the radiative flux results. Sections 3.3, 3.4, and 3.5 offer a more detailed look at the $SO_2$ emission at
height, emitted sulfate fraction, and seasonality simulation results, respectively.

### 3.1 Sulfur and BC lifetimes

One of the central factors that influences model emissions responses are the atmospheric lifetimes of BC, $SO_2$ and sulfate. In this
section we examine how reference case lifetimes vary between models as context for the analysis of perturbation responses in the
next sections. Figure 1 shows the sulfate lifetime (i.e., sulfate column burden divided by the sum of the dry and wet sulfate
deposition rates) averaged over the globe and an approximate $SO_2$ lifetime (i.e., $SO_2$ column burden divided by emission rate of
anthropogenic $SO_2$) averaged over the Northern Hemisphere (NH) land area. Lifetime was calculated differently for $SO_2$ and $SO_4$
since the additional sink terms for $SO_2$ (i.e., gas-phase and aqueous-phase oxidation (Liu et al., 2012)) were not available from the
standard output of the models. Therefore, anthropogenic $SO_2$ emission flux was used as the sole source term. Although the $SO_2$
lifetime as calculated here will be biased high since DMS and volcanic source terms were not used in the calculation (diagnostic
data was not available for all models), we focus on the value over NH land where anthropogenic emissions dominate and this
source of bias is small compared to the inter-model variation.



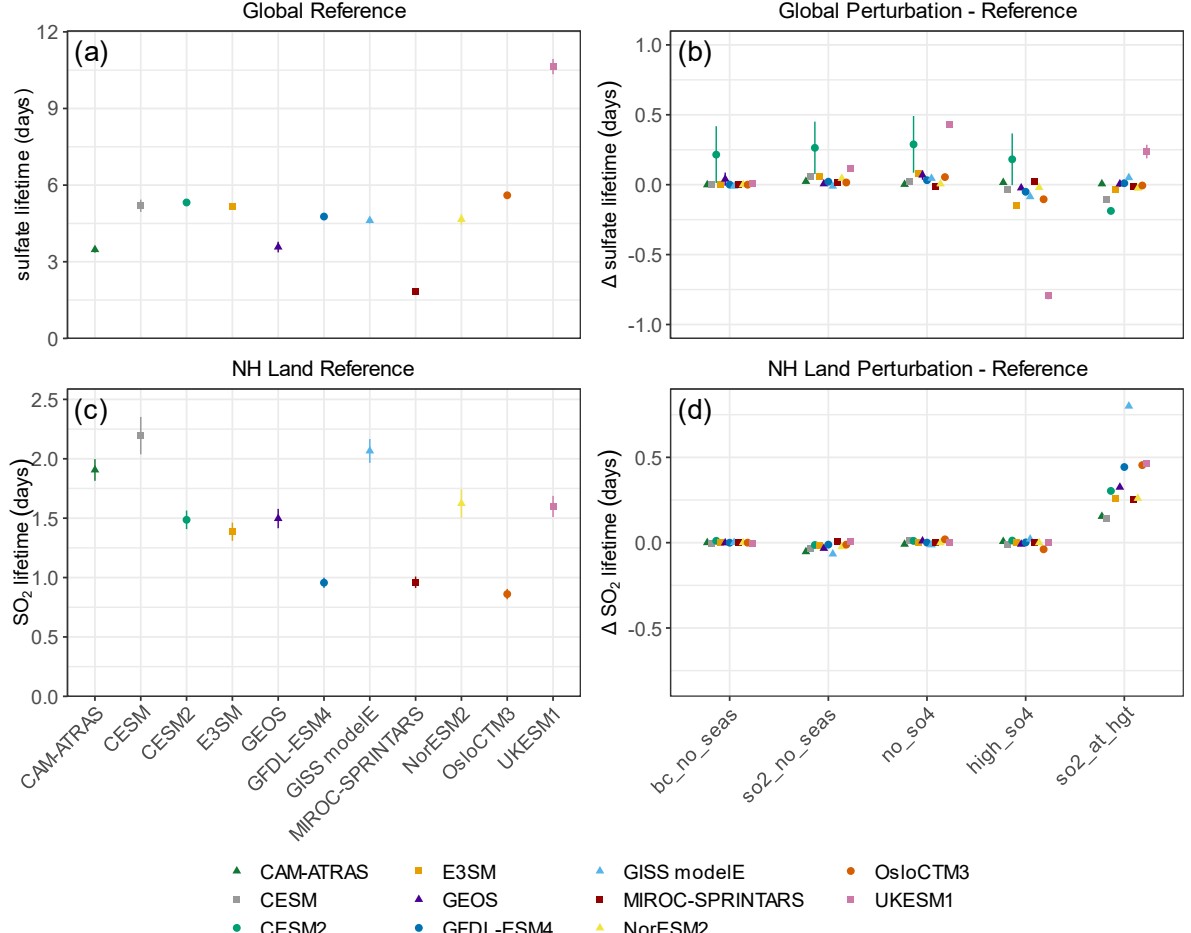

**Figure 1: (a) global sulfate lifetime (c) and Northern Hemisphere land SO₂ lifetime of the reference case model simulations, and (b, d) absolute difference between each perturbation and the reference case. All results are averaged over the years 2000 – 2004, except NorESM2 is averaged over 2001 – 2005. The error bars represent interannual variability (± 1 σ). Note that the large uncertainty bars for CESM2 sulfate lifetime is due to the high interannual variability in the sulfate column burden.**

The sulfate lifetime for the reference case in Figure 1a is 5 days on average, with a range of 3.5 – 5.6 days excluding two outlier values. The lifetime for UKESM1 was considerably higher, at 10.6 days due to the low wet deposition rate of sulfate in this version of the model. UKESM1 emits primary $SO_4$ at a relatively small diameter of 100 nm (geometrical mean) which reduces cloud droplet nucleation efficiency. The version of the model used in the current study also has a relatively high scavenging diameter (i.e., the particle diameter above which particles are removed in large-scale rain events, prescribed here as 150 nm), which increases

the number of particles that pass through clouds to reach higher altitudes and thus increases sulfate lifetime. The other outlier value was MIROC-SPRINTARS, with a sulfate lifetime of 1.8 days. In part, this low value is because this model is known to exhibit a lower sulfate lifetime in nudged simulations using reanalysis atmospheric data in which the response of precipitation tends to be excessive. It is not known if this effect exists in other models. In simulations without constraining meteorological fields, the sulfate lifetime is approximately doubled, which would be closer to the central range.

Models showed a greater relative variation, compared to that for $SO_4$, for the mean $SO_2$ lifetime of 1.5 days over NH land, as depicted in Figure 1c, with a range of 0.9 to 2.2 days. The variation in the $SO_2$ lifetime response is nearly proportional to that of $SO_2$ column burden (numerator) since the anthropogenic $SO_2$ emission rate (denominator) is very similar across models (Figure



S2). SO$_2$ lifetime was also examined over the globe (Figure S3) to compare the relative impact of DMS chemistry which could be a potential source of variation. The global mean SO$_2$ lifetime was 1.8 days and ranged from 1.3 to 2.5 days. When averaged across

all models, the global SO$_2$ lifetime is 20% greater than for NH land. The SO$_2$ column burden is 2.4 times higher over NH land but the emissions rate of anthropogenic SO$_2$ is three times higher compared to the global mean. The global SO$_2$ lifetime is lower than this value because ocean DMS flux from the oceans and volcanic emissions were not included in the calculation. Dry and wet SO$_2$ deposition (Figure S4) constitute about 70% of the total sink in NH land on average and do not have a strong correlation with SO$_2$ lifetime (i.e., poor linear relationship).

Figure 2 shows the global BC lifetime, which is the BC column burden divided by the sum of the dry and wet BC deposition (wet deposition is the dominant factor at about three times as high on average – Figure S5). In Figure 2a, the global BC lifetime is 5.6 days, with a fairly large range of 3.7 – 8 days. This global average and range are consistent with results from recent studies (Gliß et al., 2021; Lund et al., 2018b; Kristiansen et al., 2016; Samset et al., 2014). Removing BC seasonality had an impact on global BC lifetime in some models as shown in Figure 2b, with both positive and negative responses. GISS modelE and

UKESM1 both exhibited a noticeable drop in BC lifetime of 0.48 days and 0.29 days, respectively. The remaining models showed only a small increase in lifetime, with a maximum increase of 0.12 days for CESM2.

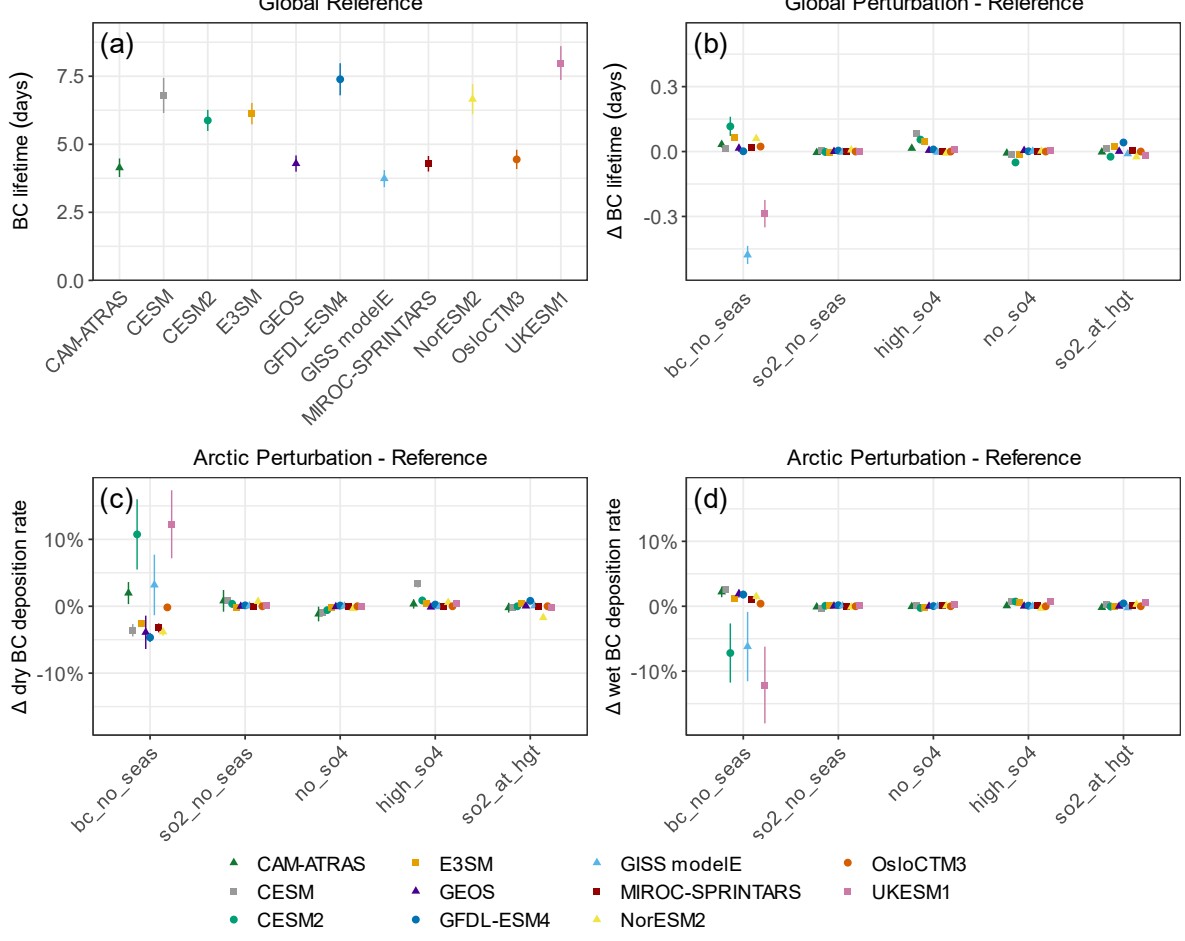



**Figure 2: (a) reference case of global BC lifetime, (b) absolute difference of global BC lifetime for each perturbation, (c) percent difference**
**of Arctic dry BC deposition rate, and (d) percent difference of Arctic wet BC deposition rate. All results are averaged over the years**
**2000 – 2004, except NorESM2 is averaged over 2001 – 2005. The uncertainty bars represent interannual variability (± 1 σ).**

### 3.2 Radiative flux

Figure 3 shows the impact of the perturbations on the radiative flux at the top of the atmosphere (perturbation experiment minus
the reference case), where a positive change denotes a heating effect, and a negative change denotes a cooling effect. Clear-sky
longwave showed a minimal response which is consistent with fixed SST experiments (since longwave would be driven largely
by surface temperature changes which are limited in fixed SST experiments). $SO_2$ emission at height consistently decreased clear-
sky shortwave flux, leading to increased cooling, with a few models showing a fairly large response (e.g., GISS modelE at -0.5 W
$m^{-2}$ and OsloCTM3 and UKESM1 at around -0.3 W $m^{-2}$). The implied cloud response exhibited a diversity of magnitude and sign.
NorESM2 had the largest change resulting from the emission height experiment, with a decrease in cloud forcing by -0.19 W $m^{-2}$.
OsloCTM3 and GISS modelE exhibited the largest increase in cloud forcing by 0.15 W $m^{-2}$ and 0.11 W $m^{-2}$, respectively. However,
OsloCTM3 only includes the direct aerosol effect and thus changes in the cloud forcing are associated with cloud response to
atmospheric adjustment rather than aerosol-cloud interactions. All remaining models showed a moderate decrease in the cloud
response.

These changes are potentially large compared to the effective radiative forcing (ERF) for aerosol-radiation interactions
(ARI) and aerosol-cloud interactions (ACI) as reported in the Intergovernmental Panel on Climate Change (IPCC) Sixth
Assessment Report – AR6. The best estimate of ERF (2019-1750) in AR6 is -0.22 W $m^{-2}$ and -0.84 W $m^{-2}$ for ARI and ACI,
respectively (Szopa et al., 2021). The changes in global mean net radiative flux we found here, for at least some models, are a
significant fraction of these values. Note that in this study we are looking at differences in radiative flux and did not formally
calculate ERF, so this is only an approximate comparison.



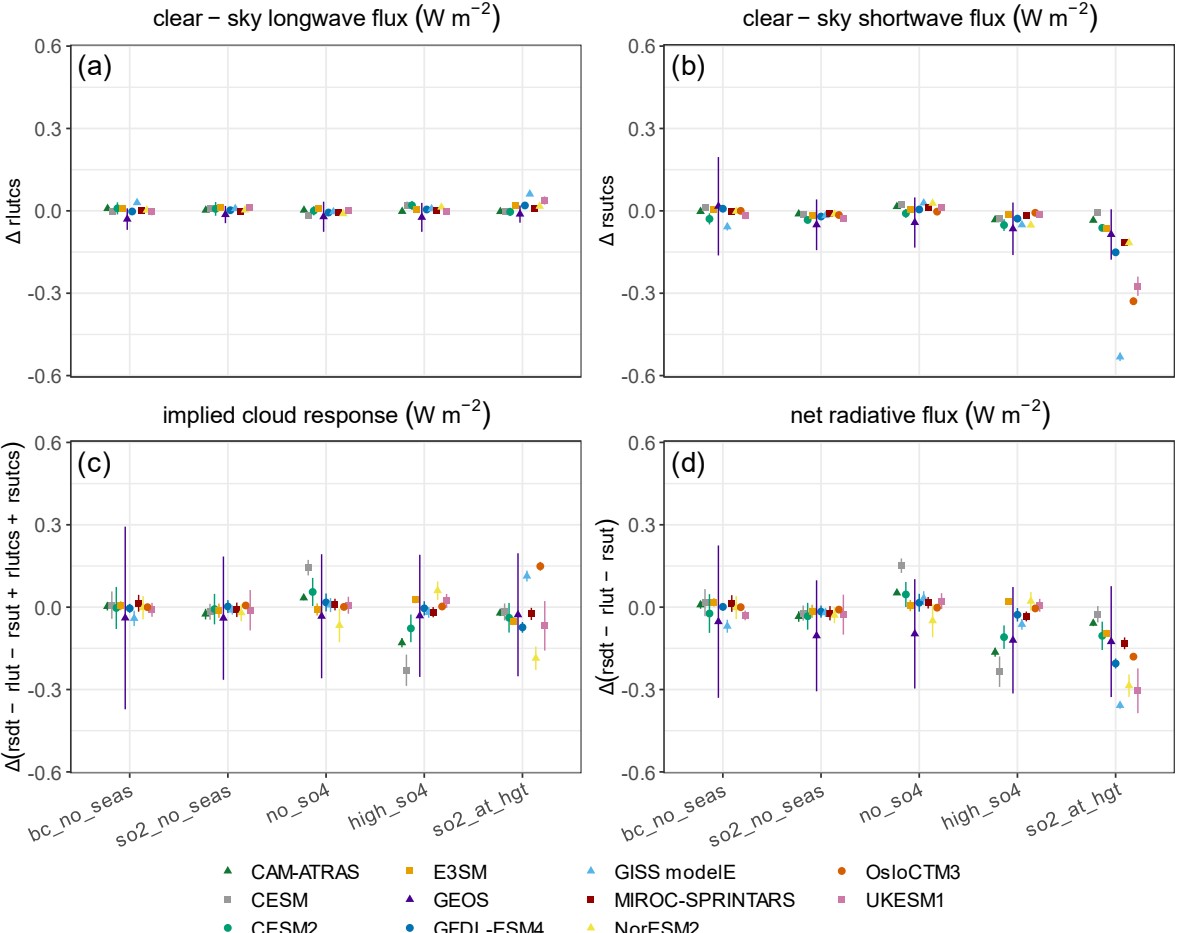

**Figure 3: Absolute difference (perturbation – reference) of global mean (a) clear-sky longwave radiative flux, (b) clear-sky shortwave radiative flux, (c) implied cloud response, which is the net forcing minus the sum of clear-sky longwave and shortwave flux, and (d) net radiative flux, averaged over the years 2000 – 2004 (NorESM2 averaged over 2001 – 2005). Interannual variability (± 1 σ) is shown as thin lines. Note that the large interannual variability in GEOS is due to a steep ramping up from 2000 to 2001 for the radiative flux variables and changes for OsloCTM3 is only due to aerosol-radiation interaction.**

### 3.3 SO₂ emission at height

The SO$_2$ emission at height results exhibited both increases and decreases in sulfate lifetime (Figure 1b). CESM and CESM2 showed a decrease and UKESM1 had an increase in lifetime. The reason for this is nuanced since emission at height not only increased sulfate dry and wet deposition, but it also increased the sulfate column burden. The signs of these two effects were consistent across all models. Therefore, an increase in both the numerator and denominator may result in either a positive or negative difference (i.e., perturbation experiment minus reference) depending on the relative magnitude of each effect. Overall, the emission height assumption had a relatively small impact on sulfate lifetime for most models.

Turning to SO$_2$ lifetimes, Figure 1d shows that emission at height consistently increases SO$_2$ lifetime over Northern Hemisphere land. The largest increase is 0.8 days in GISS modelE, with an average of 0.31 days across the rest of the models (range of 0.14 – 0.47) and a proportionate increase in SO$_2$ column burden (Figure S6). We also note that the four highest model responses (GISS modelE, UKESM1, OsloCTM3, GFDL-ESM4) all have endogenous oxidants in their model configuration. The total SO$_2$ deposition rate dropped across all models (Figure S7), with an average increase in wet SO$_2$ deposition rate of 1.5x10$^9$ kg



yr$^{-1}$ (21%) which is smaller than the average drop in dry SO$_2$ deposition rate of 1.2x10$^{10}$ kg yr$^{-1}$ (40%). Emission at height, therefore, also results in a shift from dry to wet deposition.

As the sink via deposition becomes slower (due to being further emitted from the surface), the other sink pathway (conversion to SO$_4$) becomes more important. While we do not have diagnostics available for chemical conversion, we can infer the relative importance of deposition vs chemical conversion by estimating the change in atmospheric lifetime if we assume a

constant atmospheric SO$_2$ oxidation rate. We find that the change in SO$_2$ lifetime is smaller by an average of a factor of 1.7 (range 1.3 – 2.0) than seen in the model results (Figure S8 and Table S1) if the only change was SO$_2$ deposition. This means that the SO$_2$ lifetime increase due to decreased deposition for emission at height is being significantly offset by an increase in the rate of SO$_2$ conversion to SO$_4$ through either gas-phase or aqueous-phase processes. This is also indicated in the change in sulfate burden change, which exhibits a reasonable correlation with the offset in SO$_2$ lifetime (Figure S9). In summary, we find that SO$_2$ emitted

at height results in decreased SO$_2$ deposition and an increase in oxidation to sulfate, which in turn increases the sulfate burden.

For SO$_2$ emission at height, there were small positive and negative changes in BC lifetime. The reason for these changes may be due to aerosol mixing between BC and sulfate or atmospheric adjustments.

Of the perturbations considered, the emission at height experiment had the largest impact on net flux, with impacts of up to -0.35 W m$^{-2}$ for GISS modelE and two additional models at around -0.3 W m$^{-2}$, and the remaining models ranging down to

nearly zero (Figure 3d). The range in net forcing is a combination of the range in individual forcing responses and the fact that the cloud responses have different signs. This has important implications on model calibration and tuning. For instance, OsloCTM3 and GFDL-ESM4 exhibited a similar net flux response, but the radiative flux components that contribute to the net flux differed significantly. With GFDL-ESM4, a modest change in the cloud response and clear-sky shortwave flux combined into a large change in net flux. However, for OsloCTM3 these terms were both large, but of opposite sign. This diversity of responses is an

indicator of the significant uncertainty in the underlying mechanisms driving aerosol forcing across models.

Examining the SO$_2$ emission height results in more detail, we find a strong relationship between the change in clear-sky shortwave forcing and change in sulfate column burden (Figure 4a). The change in sulfate column burden ranges from 0 to 25% (Figure S10) relative to the reference case. With a couple of outliers, this relationship is remarkably linear across the models given the many factors that could potentially influence this relationship such as sulfate particle size distribution, optical properties, and

mixing treatment, although we note that a number of models represented here have aerosol schemes related to the CESM family of models (Liu et al., 2012, 2016). GISS modelE, given the column burden change, has a stronger relative shortwave response compared to the other models, potentially due to new particle formation (i.e., the formation of Aitken size sulfate particles from binary nucleation) and the interaction with nitrate aerosol formation processes, as well as a stronger height dependence for sulfate production. The sulfate column burden is driven by an increase in SO$_2$ column burden, since emitting SO$_2$ at height (Figure S11)

consistently increases SO$_2$ lifetime (Figure 1d). Although the sulfate lifetime did not show a consistent change due to the emission at height experiment (Figure 1b), there was an increase in sulfate in the atmosphere (Figure S12) due to the increase in SO$_2$ column burden, as illustrated in Figure 4b. This is a fairly linear relationship, with the exception of the OsloCTM3 model, which showed a stronger response to SO$_2$ column burden. The reasons for this different response were not clear but is perhaps due to nonlinearity in lifetime changes with height.

Model vertical resolution was another factor that has an impact on these results, particularly for the SO$_2$ emission height experiment. Figure 4c shows that with increasing model vertical resolution (i.e., decreasing layer thickness) the model response increased, except for GISS modelE. The relatively coarse vertical model resolution in GISS modelE introduces stronger sensitivities towards the collocation of aerosol and cloud layers, and therefore strongly impacts aerosol-cloud interactions, such as in-cloud aqueous chemistry rates, aerosol activation, and wet removal. We observe a cluster of relatively high- and low-resolution





models. The high-resolution models have a stronger clear-sky shortwave flux response in general, but still with variation across
this subset of models. Two of the models with a relatively high response (i.e., OsloCTM3 and UKESM1) are higher resolution
models. In contrast, E3SM had a lower sensitivity compared to the other high-resolution models, as also shown by Figure 4d which
illustrates a fairly linear relationship between sulfate column burden change for models with more than two layers below 400 m
excluding E3SM. Although E3SM has the same number of layers below 400 m as UKESM1, it had a notably smaller sulfate burden

response, likely due to a difference in the treatment of sub-grid vertical mixing and transport. Differences in $SO_2$ lifetime do not
appear to explain the shortwave response among the high-resolution models (Figure S13) since the difference in OsloCTM3 and
UKESM1 lifetime is relatively large (0.75 days).

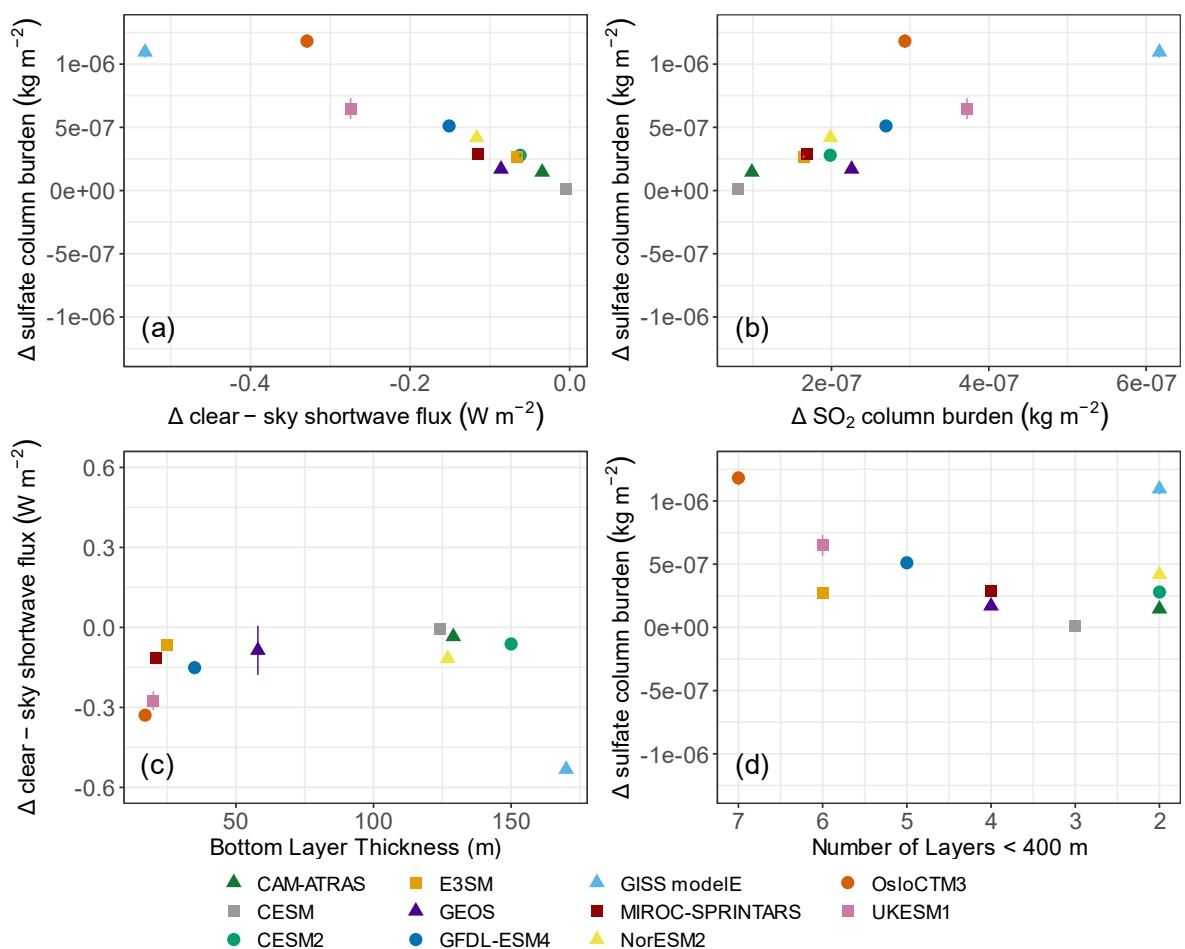

**Figure 4: Impact of $SO_2$ emissions at height on the relationship between (a) sulfate column burden vs clear-sky shortwave flux changes, (b) sulfate column burden vs $SO_2$ column burden changes, (c) clear-sky shortwave flux change vs bottom model layer thickness, and (d) sulfate column burden change vs number of model layers below 400 m.**

The emission height protocol described in Section 2.3.1, which distributed emissions to 200 – 400 m above land surface, falls

below the average model planetary boundary layer height (PBLH) of 637 m over NH land as shown in Figure 5. The average
PBLH over NH land has four models clustered together at around 650 m, although the full range across models is 283 – 947 m.





While the emission height is lower than the average PBLH, it is important to consider that the PBLH can be considerably lower during the night, for example around 250 m during the night compared to 800 m during the day (Svensson et al., 2011). Since there is more stratification of PBLH during night, emission height can make a bigger difference, but it is not clear how the PBLH

interacts with mixing schemes in the models and how they behave diurnally (Maier et al., 2022). In the context of the current study, this suggests that some of the emissions would be above the boundary layer during the night, which may explain why emission height has a significant impact on some model results. While there was no apparent correlation between average PBLH and the emission height results, we did not have diurnal PBLH information from the models.

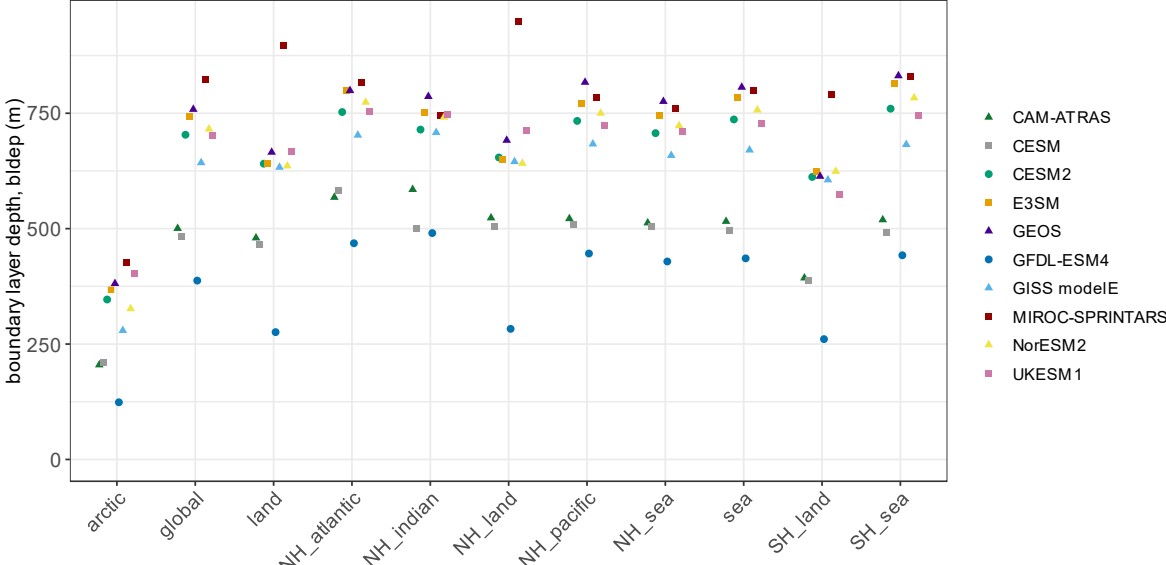


**Figure 5: Annual average model boundary layer depth across regions.**

The $SO_2$ emission height experiment also had a substantial impact on the surface concentrations of $SO_2$, with some of the highest relative changes for any variable examined (Figure 6a). Globally averaged $SO_2$ surface concentrations dropped with emission at

height, by an average drop of 39% relative to the reference case and a range of 9% – 59%. In terms of regional responses, the $SO_2$ surface concentration dropped more significantly over land (46% on average) compared to over the oceans (6% on average), as shown in Figure S14.

The $SO_2$ emission height had the opposite effect on the surface concentration of $SO_4$ than on $SO_2$, with an average increase of 10% in global surface $SO_4$ concentration and ranging from 1% to 23% (Figure 6b). The average model surface sulfate

concentration increased by a similar amount over land (10%) and over oceans (11%), as shown in Figure S15. Given that there is little change in sulfate lifetime (Figure 1b), the increased surface sulfate appears to be the result of increased conversion of $SO_2$ to sulfate due to decreased dry deposition of $SO_2$.

A strong relationship between column burden change and surface concentration change of $SO_4$ is observed in the emission height experiment (Figure S16). There is, however, not a consistent relationship across models between changes in $SO_2$ column

and surface concentrations. This is due in large part to the shorter $SO_2$ lifetime (Figure 1c) that results in more variation in the relationship between $SO_2$ surface and column changes. Also, since $SO_2$ is injected directly into the bottom model layer as opposed




to a higher layer, we would expect a larger change in surface concentrations given the same column burden. This is evident in Figure S17, where the higher resolution models (i.e., models with smaller bottom layer thickness) are shown to have a larger drop in $SO_2$ surface concentration in the emission height experiment.


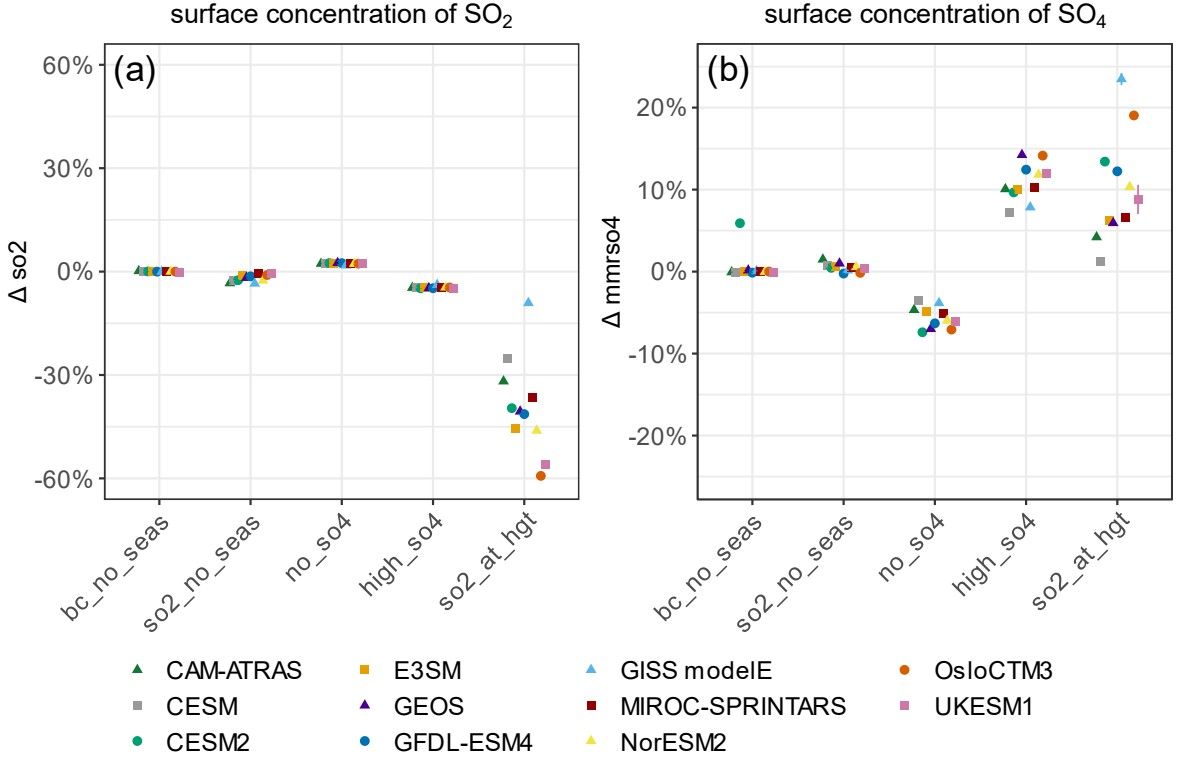

**Figure 6: Global percent difference (perturbation – reference)/reference of (a) surface concentration of SO₂ and (b) surface concentration of SO₄. All results are averaged over the years 2000 – 2004, except NorESM2 is averaged over 2001 – 2005. The error bars represent inter-annual variability (± 1 σ).**

**3.4 Emitted sulfate fraction**

When the sulfate fraction of emissions is increased sulfate lifetime decreases (and conversely with no S emitted as $SO_4$), although the effect is small for some models (Figure 1b). This result is explained by changes in sulfate deposition, which increases with a higher emitted sulfate fraction, while the sulfate column burden showed minimal changes. Note that this is in the baseline experimental setup with all emissions injected to the lowest model layer, where more $SO_2$ emitted as $SO_4$ can be more readily lost

to dry deposition, although the strength of this effect varies by model. This may be dependent on depth of the lowest model layer (i.e., change in sulfate deposition due to a higher sulfate fraction generally increases with layer thickness, as shown in Figure S18).

Sulfate emission fraction also consistently changed the BC lifetime in a couple of models. CESM2 showed a slight increase (less than 0.1 days) in BC lifetime in the no sulfate fraction experiment and a decrease in lifetime by a similar magnitude in the high sulfate fraction experiment. CESM and E3SM also showed an increase in BC lifetime for no sulfate but a smaller

decrease in lifetime for a high sulfate fraction.

Increasing the sulfate emission fraction consistently decreased clear-sky shortwave flux slightly, but the largest changes were to cloud response, again with both positive and negative responses in different models. The responses to sulfate fraction



perturbations may be a reflection of the cloud cover change (Figure S19), which is generally positive (i.e., an increase in cloud cover) for high sulfate fraction and negative for no sulfate upon emission. However, the NorESM2 cloud response had the opposite

sign compared to the other models for the sulfate fraction experiments. This appears to be due to a response in ice water path (Figure S20) which shows a relatively strong response for NorESM2, with an increase for the high sulfate experiment and decrease for the no sulfate experiment.

       The high sulfate fraction experiment yielded a decrease in the net radiative flux and cloud response, averaging -0.064 W $m^{-2}$ and -0.036 W $m^{-2}$ across the models, respectively. This is consistent with the notion that sulfate aerosols can act as CCN and

affect cloud formation, as well as having a cooling effect on the climate (Takemura, 2020). The experiment with no sulfate emission fraction exhibited opposite signs in net radiative flux and cloud response for most models, with an average of 0.018 W $m^{-2}$ and 0.015 W $m^{-2}$ across models, respectively. This experiment also shows a decrease in cloud cover for nearly all models (Figure S19).

       Furthermore, the assumption on primary sulfate emission fraction had an impact on global surface $SO_4$ concentration. As illustrated in Figure 6b, the high sulfate fraction experiment yielded an average increase in surface concentration of about 11% and

the "no sulfate emission" experiment resulted in a drop of about 6%.

### 3.5 Seasonality

The "no $SO_2$ seasonality" experiment showed a consistent increase in sulfate lifetime of 0.06 days averaged over all models. The underlying cause of this change can be attributed to the difference in sulfur emissions between the Northern and Southern Hemispheres. The Northern Hemisphere generally experiences more seasonal emissions changes due to energy consumption for

heating in the winter months. The increase in total sulfate deposition rate in the Northern and Southern Hemispheres with no emission seasonality, averaged across all models, is $1.04 \times 10^{13}$ kg $m^{-2}$ $s^{-1}$ and $7.72 \times 10^{15}$ kg $m^{-2}$ $s^{-1}$, respectively (Figure S21). This is further corroborated in Figure 7, which shows a higher sulfate lifetime in the Northern Hemisphere due to $SO_2$ seasonality with the exception of CESM2, which is inconclusive.





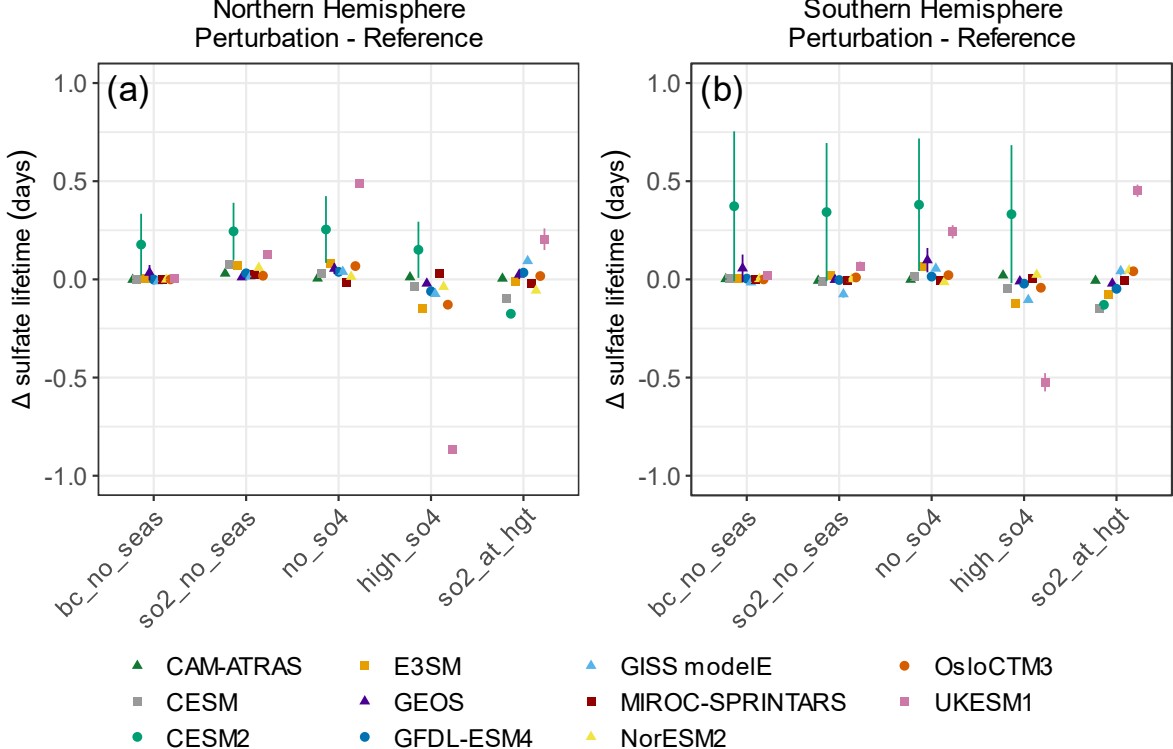


**Figure 7: Absolute change in sulfate lifetime averaged over the (a) Northern Hemisphere and the (b) Southern Hemisphere. The error bars represent interannual variability (± 1 σ).**

Previous studies have shown that the Arctic BC concentration, deposition, and source attributions have a strong seasonality (Matsui
et al., 2022; Ren et al., 2020; Wang et al., 2014, 2013; Stohl et al., 2013). We also find that BC deposition rates in the Arctic are
sensitive to BC seasonality, although not consistently across models. In the reference case, based on CMIP6 historical data, BC
emissions in the Arctic were at a maximum of 2.9 kt in January and a minimum of 1.9 kt from June to August, as shown in Figure
8a. The global BC emissions in Figure 8b also show a maximum and minimum during winter and summer respectively, although
the degree of seasonal variation is not as distinct. The impact of seasonality on deposition is not consistent between models, with
one set of models showing an increase in dry deposition when emission seasonality was removed, while another set shows the
opposite behavior, although at a lower magnitude (Figure 2c). The opposite behavior is seen for wet deposition except for CAM-
ATRAS (Figure 2d). In the CAM-ATRAS model, seasonality increases BC transport to the Arctic during the winter, which may
increase the annual-mean BC concentration and dry/wet deposition in the Arctic. The simulated seasonal variability of precipitation
is a potential driver of the differences observed between models, as well as BC transport and height. We note that the interannual
variability for models with an increase in dry BC deposition was much more prominent than for those models that showed a
decrease.



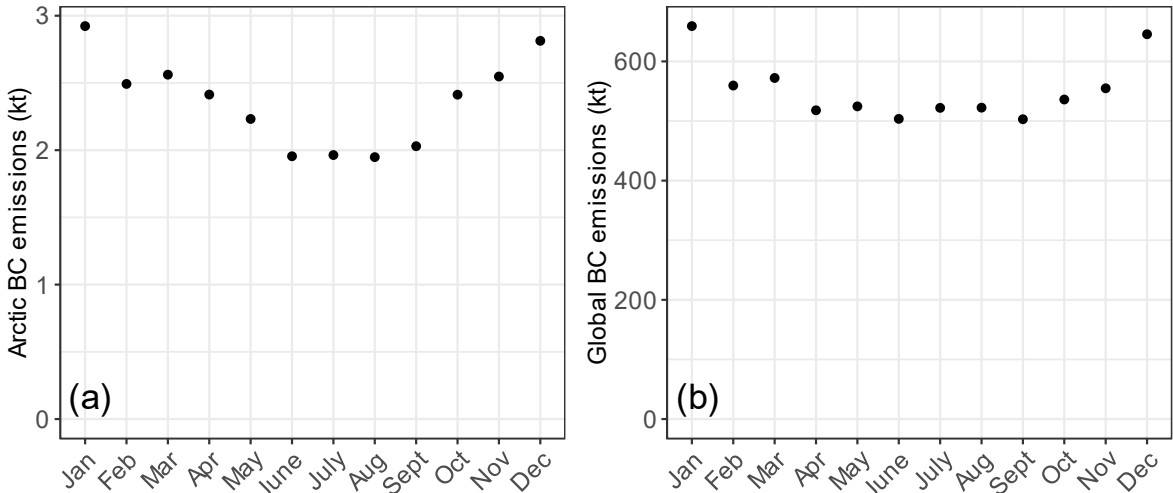

**Figure 8: Reference case (a) Arctic BC emissions (> 66° N) and (b) global BC emissions based on monthly CMIP6 data for 2004.**


$SO_2$ seasonality did not have a large impact on any of the forcing metrics. BC seasonality had a slightly larger impact, particularly for GISS modelE, but the magnitude of the effect was small (Figure 3).

### 4 Conclusions

This study explored the sensitivity of 11 climate-aerosol and chemical transport models to four emission characteristics: $SO_2$
emission height, $SO_2$ seasonality, BC seasonality, and the fraction of $SO_2$ assumed to be $SO_4$ upon emission. Each perturbation experiment used atmosphere-only model simulations with specified sea surface temperatures and nudged winds, running for a five-year period following one year spin-up. Of the perturbations examined in this study, the assumed height of $SO_2$ injection had the largest overall impacts, particularly on net radiative flux (maximum absolute difference of -0.35 W m$^{-2}$), but also on $SO_2$ lifetime over NH land (maximum absolute difference of 0.8 days), surface $SO_2$ concentration (up to 59% drop), and surface sulfate
concentration (up to 23% increase). The sulfate emission fraction had a nontrivial impact in some models, particularly for net radiative forcing and surface $SO_4$ concentration. $SO_2$ and BC seasonality did not have a substantial impact on the global annual mean simulation results. However, BC seasonality had a slightly larger impact on net radiative forcing and had a significant effect on BC deposition in the Arctic, where we observed both positive and negative changes for both dry and wet deposition.

In general, the assumptions on emission height and $SO_4$ fraction are a "hidden" source of inter-model variability because
models have made different assumptions about these parameters. This is in addition to differences in model structure such as aerosol microphysical parameterizations. As demonstrated here, this unquantified source of differences may have a large impact on model results. Therefore, potential modifications or new datasets are needed for these parameters to both improve model results and remove a source of inter-model difference. Five of the models used here assume all anthropogenic emissions  are injected into the lowest model layer in their default set-up. This will result in a bias in model results compared to reality for the bulk of $SO_2$
emissions. Three of the models inject emissions either at 100 m or a higher level (100 – 300 m) for industrial and power generation sectors, which will still be an underestimate of injection height for some large sources (Akingunola et al., 2018). There was more uniformity in the fraction in $SO_4$ fraction, with most models assuming 2.5% of $SO_2$ is emitted as sulfate.





Assumptions on emission height, and to a lesser extent $SO_4$ fraction, can have a very large impact on surface concentration values in the models. Evaluating model results by comparing with surface observations, particularly for $SO_2$, will also be impacted

by these assumptions. When evaluating models against observations the sensitivities explored in this work can be a potential source of bias. These issues also apply to satellite-based estimates which generally incorporate assumptions about vertical distributions. For example, the Ozone Monitoring Instrument (OMI) aboard NASA's Aura satellite detects $SO_2$ signals from anthropogenic sources (Fioletov et al., 2011) and has been compared with simulations by global models (Qu et al., 2019). These issues will be particularly large for satellite data products with more limited sensitivity to concentrations near the surface.

We find a large variation in atmospheric lifetime across models for $SO_2$, $SO_4$, and BC, particularly for $SO_2$. The underlying drivers of this variation also likely drive some of the variation in results seen in the perturbation experiments. Better observational constraints on processes that influence aerosol lifetime (e.g., deposition, aerosol microphysical processes such as nucleation, coagulation, gas-to-particle conversion, ageing (for BC)) are needed to improve model physics and chemistry. Samset et al. (2014) used aircraft-based measurements of BC concentration to constrain BC radiative forcing and atmospheric lifetime in global aerosol-

climate models, and this led to a reduction of 25% in anthropogenic BC direct radiative forcing in remote areas relative to default model values. The UKESM1 model has recently incorporated updates to the aerosol removal processes, specifically through convective plume scavenging, nucleation scavenging, and dry deposition and sedimentation (Mulcahy et al., 2020). As part of a study to reduce uncertainty in the UKESM1 model through observational constraint, Regayre et al. (2022) show that dry deposition is one of the largest causes of uncertainty in aerosol forcing that remains largely unconstrained, even when other causes of

uncertainty are tightly constrained. Better constraining $SO_2$ chemistry in atmospheric models remains an important research goal for the community.

Model vertical resolution was found to have a large impact on the $SO_2$ emission height experiment, with a higher vertical resolution corresponding to a stronger clear-sky shortwave flux response. However, there was still a relatively large diversity in response among the high-resolution models. E3SM demonstrated a weaker sensitivity to clear-sky shortwave flux and sulfate

column burden compared to the other high-resolution models (i.e., OsloCTM3 and UKESM1), so there are other underlying factors at work. We note that one of the last simulations done by most of the participating modeling groups was the emission at height simulation, as this required, in some cases, altering either model setup, data pre-processing, or internal model code. This points to the importance of carefully considering the best approach to incorporating these effects into global models.

These results imply a need to assure that anthropogenic emission injection height is accurately and consistently

represented in global models. This is in addition to considering the impact of biomass burning injection height which already has significant research (Veira et al., 2015; Paugam et al., 2016; Zhu et al., 2018). Collecting consistent data on emission stack height is one challenge, although such data often exist regionally (e.g., in USA, Europe). In the context of models, we need the effective injection height, which is stack height plus plume rise, where plume rise is dependent on both stack characteristics, particularly effluent temperature, and meteorological conditions (wind speed, temperature, and presence of any inversion layers). The effective

injection height will also depend on the diurnal cycle of meteorology, PBLH and stability. This points to the difficulty of providing accurate information on effective injection height globally. One option might be to implement plume rise parameterizations in global models. Another option is to collect information on the average amount of plume rise estimated in regional models to inform guidance for global models. Note that as model vertical resolution increases, the effects we found here become more important, and some solutions (such as plume rise parameterizations) may become more practical, or perhaps even necessary, for some model

applications. At minimum models should clearly report their emission injection height assumptions and model intercomparison exercises should consider if standardized guidance should be provided.





The emissions at height perturbation experiment, in particular, is a novel diagnostic of the systemic response of a model to a fundamental change in emission characteristics. As discussed in the main text, this variety of model responses seen from this experiment expose substantial variation, and therefore, uncertainty in aerosol dynamics and forcing responses across models.

We note further that the models used in these studies ignore $SO_3$ emissions emitted at stacks, which may impact results. This is important since $SO_3$ in the atmosphere can potentially form sulfuric acid, which in turn can nucleate or condense to existing particles. Coal plants in China with pollution controls in place have been found to emit up to 40% of their sulfur in the form of $SO_3$ (Wu et al., 2020). Other work seems to support the notion that the ratio of $SO_3$ to $SO_2$ increases as controls strengthen. Mylläri et al. (2016) establish that flue-gas cleaning technologies greatly reduce $SO_2$ concentration, and they further suggest that $SO_3$ may

exist in the plume and can increase the probability of aerosol formation.

   Current global inventory data is not necessarily consistent in accounting for emissions of different sulfur species (e.g., $SO_3$ and $SO_2$ gas, and filterable and condensable $SO_4$ particles). Bottom-up mass balance approaches, which rely on data on fuel sulfur content, are implicitly reporting all sulfur-containing species as $SO_2$. Inventories that rely on measurement data, such as data from stack concentration monitoring systems, are reporting $SO_2$ emissions only, which may lead to "missing" sulfate emissions

when this data is used in models (Ding et al., 2021). This points to a need to harmonize how sulfur-containing emission species are reported and how this data is interpreted within modeling systems.

**Data availability**. The input emission data files supplied to the modeling groups have been archived at https://doi.org/10.25584/DataHub/1769948. A full set of global and regional time series results and diagnostic graphics are

available here: https://github.com/JGCRI/Emissions-MIP_Data and have been archived here: https://doi.org/10.5281/zenodo.7765075.

**Supplementary Material**. A supplementary file Emissions-MIP_Phase1a_Supplement.pdf provides additional figures and tables. Supplementary file Emissions-MIP Experimental Protocol - v1b.xlsx provides the full experimental protocol.


**Author contributions**. HA: formal analysis, software, visualization, writing – original draft, writing – review and editing; HW: conceptualization, investigation, methodology, writing – review and editing; JW: investigation; MW: investigation, writing – review and editing; SJS: conceptualization, formal analysis, funding acquisition, project administration, investigation, writing – original draft, writing – review and editing; SB: conceptualization, investigation, methodology, writing – review and editing; HS:

software; DO: investigation, methodology, writing – review and editing; GM: methodology, writing – review and editing; HM: investigation, methodology, writing – review and editing; HB: investigation; JL: investigation, methodology; KC: methodology; LH: methodology, writing – review and editing; LR: investigation, methodology, writing – review and editing; MC: methodology; MS: methodology; RBS: investigation, writing – review and editing; TT: investigation, methodology, writing – review and editing; VN: investigation, methodology, writing – review and editing.


**Competing interests**. At least one of the (co-)authors is a member of the editorial board of *Atmospheric Chemistry and Physics*. The peer-review process was guided by an independent editor, and the authors also have no other competing interests to declare.

**Acknowledgments**. This research at PNNL was supported by the US Department of Energy, Office of Science, Office of

Biological and Environmental Research, Earth System Model Development Program Area of the Earth and Environment Systems Modeling Program. The OsloCTM3 model group acknowledge support from the Research Council of Norway (grant no. 314997).



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
