# Peer review of "The Emissions Model Intercomparison Project (Emissions-MIP): quantifying model sensitivity to emission characteristics"

_EGUsphere, 2023_

## Author Comment (AC1)

**Response to Reviewers**

We thank the two reviewers for their insightful comments, which has helped us to refine and improve the quality of our discussion. We have addressed each comment in detail below. Our responses are in blue, with specific changes to the text highlighted in *blue italics*. All line numbers in this document correspond to the line numbers in the updated (tracked-changes) version of the manuscript.

**RC1: 'Comment on egusphere-2023-604, useful but some parts too short', Anonymous Referee #1, 16 Jun 2023**

By applying 11 models the paper shows that calculated radiative forcing by tropospheric sulfate and black carbon aerosol is very sensitive to the altitude distribution of the emissions, and also in less extent, to the share of sulfate in the emissions and simplifications in the models.

It should be published in ACP or maybe GMD after minor revisions.

We thank the reviewer for their thorough and insightful comments. We have incorporated the suggestions you have made and added more details to improve the clarity of our discussions. We have responded to each comment below.

Line 23: Define "BC" as "black carbon particles" where it occurs first.

We have added this definition to the text.

Line 24 and later: Don't use "$SO_4$" as abbreviation for sulfate without definition, at least not in the abstract. I suppose everywhere the particulate phase is meant and not gaseous $H_2SO_4$.

We thank the reviewer for pointing this out. We have added to the text that it is the particulate phase which we refer to.

Line 24 – *"and the assumed fraction of SO2 emissions that is injected into the atmosphere as particulate phase sulfate (SO4) in 11 climate and"*

We have also realized that for some of the models (i.e., E3SM, CESM, MIROC-SPRINTARS, NorESM2), the SO4 output is reported as sulfuric acid (H2SO4) or ammonium sulfate (NH4HSO4), so we have updated Figures 5, S10, S13, S17, S19, S22 with the corrected values to represent mass of SO4.

Table 1 and corresponding text in section 2.1: More information should be provided on "Endogenous oxidants, Endogenous DMS emissions". Which oxidants? For DMS effects on $SO_2$ it matters if methylsulfonic acid as oxidation product is considered or not.

We thank the reviewer for their suggestion. For roughly half the models, the oxidants are not interactive, so that does not impact our results. For the models where the oxidants are interactive, we have confirmed with the coauthors that the references listed do provide this information. Each model has different chemistry, number of species and oxidants considered. These details are mentioned in the

references provided by each model. Further, in response to the comment below, we have added in the supplement a detailed description of the aerosol representation for each model, which also mentions the coupling to chemistry where relevant.

Characteristics of the aerosol module important for the optical properties and hygroscopicity should be mentioned like e.g. interaction between BC and sulfate (aging, coating or external mixture), modal or sectional size distribution. This is important regarding the discussion in the conclusions.

We agree with the reviewer that additional details regarding the aerosol module characteristics are important to include. We have added a table with these details to the supplement (Table S2).

Line 141ff: Refer to Table 2. From the discussion before it would have been better also to include emissions above 400m (discussed in conclusions).

Table 2 is now referenced.

Line 163: Gaseous? Sulfate precursor?

In this sentence we are simply indicating that measurements have indicated that SO3 is emitted and is measured as indicated in the references, presumably in gaseous form. Whether it is a sulfate precursor or not is an issue for the aerosol schemes, not for the chemistry schemes. We are not aware that any current global models treat this emission explicitly, but instead any SO3 emissions are implicitly treated as some combination of SO2 and SO4. In situations where SO3 is a significant component of primary emissions, how this should be treated in global models should be considered in future work.

Figures 1 and S3: The selection of frames appears to be incomplete. In the supplement at least also the lifetime of sulfate for NH land should be provided.

We have added this figure to the supplement (Figure S2).

Line 241f: This might be shortened because of a similar statement in line 215f.

Good point, thank you. We have removed the sentence since it is already conveyed in line 215.

Figure 2 and line 245ff: In the supplement should be a figure corresponding to panel a and b for Arctic BC lifetime which has to be mentioned in text.

We thank the reviewer for their suggestion. We have generated lifetime metrics for large hemispheric regions, and while there is some transport error likely involved, these regions are large enough that the results are likely reasonable. For small regions such as the Arctic, we do not feel that a lifetime metric would be robust since we do not have diagnostics on BC transport from the models.

Line 269: Here also the data of panel d of Figure 3 should be discussed since they are in the abstract and

conclusions (refer at least to line 308ff, rearrange?).

*We have added a statement here to clarify that more details on radiative flux will be discussed. Section 3.2 provides an overview, and sections 3.3, 3.4, and 3.5 go into more detail with respect to the specific perturbation experiments.*

*Line 276 – Further details on radiative flux are discussed in the following sections as they pertain to the specific perturbation experiments.*

Line 272: There appears to be a number missing (-1.06?) or the language is confusing. It would be also more convenient for the reader to provide the relating equation(s) of the forcing quantities (e.g. ERF=ARI+ACI) in section 3.2 for clarity. Is the comparison present to preindustrial only for sulfate plus BC? In the reference (Szopa et al., 2021) it includes also OC and maybe ammonia. Caution, there is an inconsistency between their Fig. 6.12 and the numbers in its caption. It is also somehow inconsistent with Fig. 7.5 (other time for present) but not Fig. 7.6 in Forster et al. (2021), which is in the reference list but not cited. The paragraph or the whole section 3.2 has to be revised regarding the relations between the quantities shown in Figure 3 and the ones in the IPCC-report.

*We thank the reviewer for pointing out the need for some clarifications. We have updated this sentence as follows.*

*Line 278 – These changes are potentially large compared to the effective radiative forcing (ERF), which is the sum of aerosol-radiation interactions (ARI) and aerosol-cloud interactions (ACI), as reported in the Intergovernmental Panel on Climate Change (IPCC) Sixth Assessment Report – AR6 (Forster et al., 2021). The best estimate of ERF (2019 relative to 1750) in AR6 is -1.06 W m$^{-2}$ (i.e., ERF = ARI + ACI, where ARI and ACI are -0.22 W m$^{-2}$ and -0.84 W m$^{-2}$, respectively (Szopa et al., 2021)).*

*Upon closer inspection of Fig. 6.12 from Szopa et al. (2021) it appears that the sum of the forcing for aerosol-cloud is about -0.85 W m$^{-2}$ and for aerosol-radiation about -0.18 W m$^{-2}$, which is reasonably close to the numbers in its caption. It also appears that the reason for the difference between these values and Fig. 7.5 in Forster et al. (2021) is that the latter shows the aerosol effective radiative forcing from 1750 to 2014, rather than to 2019. However, we have emphasized in the text that we can only compare general magnitudes and we cannot calculate ERF as part of this project.*

Line 281: Is there a problem with too short spinup for GEOS?

*Thank you for pointing this out. We have consulted with our coauthors and found that the GEOS model had a 3-month spinup which appears to be short for radiation fields. Therefore, we have removed the values for the year 2000 for GEOS. We have updated Figures 3 and 5.*

Line 488 or later: Add something like "This implies also that emission inventories should contain emissions at different altitudes typical for the source categories (power, industry, road traffic, shipping...)".

Thank you for the comment. We have added this statement as noted.

**Technical corrections**

Table 1, column 3: Add "latitute x longitude" after "Resolution". Is the order for GISS consistent? "MATRIX" and the corresponding reference should be connected by a superscript or symbol.

Thank you for pointing out the issue with GISS. This has been fixed and the other changes made.

Lines 223, 383, 421: Better skip "error" here.

Changed.

Line 282: Cite Table 3 for the definition of the variables.

Reference has been added.

Line 361: Refer to Figure S1 for definition of regions.

Reference has been added.

Line 397: Refer to Figure 3c.

Reference has been added.

Line 707ff: Put the Forster reference to alphabetical order. The citation of the book is inconsistent with line 775ff.

References have been updated.

**RC2: 'Comment on egusphere-2023-604', Anonymous Referee #2, 22 Jun 2023**

In this study, the authors investigated the sensitivity of model outcomes to various factors, including the assumed height of SO2 injection, seasonality of SO2 and BC emissions, and the assumed fraction of SO2 emissions transformed into SO4 in 11 climate and chemistry models. The results revealed significant variations in the atmospheric lifetime of SO2, SO4, and BC across different models, underlining the importance of accurately accounting for anthropogenic emission injection height and SO4 emission fraction in global models. This paper is rightfully within the scope of ACP, however, I noticed several issues in this manuscript that cannot be passed over to be published. I suggested that the authors should consider the following comments.

We thank the reviewer for their helpful comments and suggestions. We have removed duplications, improved organization, and added more details and a figure to strengthen our discussion. We have responded to each comment below.

In Section 3.1 and 3.2, apart from showing results of reference scenario, authors also display perturbation results. These results have some similarities with the following contents in Section 3.3, 3.4 and 3.5. Therefore, I suggest reorganizing these results to enhance clarity and coherence. Effects of perturbations on BC are discussed in Section 3.4 and 3.5, but missed in Section 3.3, authors had better to add related results to make it comprehensive. The Conclusions section can be more concise to improve the readability.

We thank the reviewer for the suggestion to modify the order of the results. Sections 3.1 and 3.2 provide an overview of the key model diagnostics (i.e., sulfur and BC lifetimes and radiative flux). We dedicated sections 3.3, 3.4, and 3.5 to each of the perturbations so that we can discuss specific details of the sulfur cycle diagnostics and forcing results as they pertain to each perturbation. This was a deliberate decision so that it would be easier to see which uncertainty factor (injection height, sulfate fraction, or seasonality) stand out for specific metrics. It also parallels the order of sections under Section 2.3 (Overview of Perturbation Assumptions). Section 3.3 is the perturbation for SO2 at height which has minimal impact on BC, so it is not discussed.

Furthermore, it would be beneficial if the authors incorporated observations to evaluate the performance of different models under the reference scenario.

We appreciate the suggestion by the reviewer to compare model performance with observations. However, evaluating the models with respect to observations is a significant task and we refer to the individual model papers for that. Note that we have made our results publicly available in case anyone in the community would like to further evaluate the results.

Additionally, including figures depicting the global distribution of air pollutants would be valuable in illustrating the spatial patterns of SO2, SO4, and BC influenced by various emission characteristics. These additions would enhance the study's comprehensiveness and provide a visual representation of the findings.

We agree with the reviewer that adding some visual representation of the findings will be beneficial. We have decided to add global maps (Figure 4) showing the net radiative flux for the SO2 emission at height perturbation experiment (i.e., the difference from the reference case), since this is the largest sensitivity found in our results and, therefore, conveys the significance of this sensitivity for regional forcing.

Section 2.3: This section can be combined into Introduction. There are too many repetitions with contents in Introduction.

We thank the reviewer for pointing out some duplication in the introduction and Section 2.3. We have removed some instances where we found that a statement in Section 2.3 was similar to one in the introduction. Section 2.3 is intended to delve more deeply into the literature pertaining to each perturbation. Keeping it separate will avoid a lengthy introduction and will make it easier for the reader to follow the text.

Line 213: Are there any supporting materials for the method?

We have reorganized the text to clarify the discussion on calculating SO2 lifetime vs sulfate lifetime as follows.

Line 215 - *Sulfate lifetime was calculated as sulfate column burden divided by the sum of the dry and wet sulfate deposition rates. SO2 lifetime was calculated as SO2 column burden divided by the emission rate of anthropogenic SO2. The source term (i.e., anthropogenic SO2 emission flux) was used for SO2 lifetime since not all sink terms for SO2 (i.e., gas-phase and aqueous-phase oxidation (Liu et al., 2012)) were available from the standard output of the models.*

The reference included here (Liu et al., 2012) shows a breakdown of the sources and sinks for SO2 in Table 2.

Line 216: What is DMS short for?

Thank you for pointing this out, the text has been modified to clarify that DMS is Dimethyl sulfide.

Line 259: Why did you investigate the perturbations on the radiative flux at the top of the atmosphere? You need to clarify whether it is upwelling or downwelling radiative flux.

We examine the impact of these perturbations on radiative flux at top of atmosphere because this is a key term used in calculations for the Earth's energy budget. We have also added a footnote (page 10) to clarify that we are referring to upwelling radiative flux.

Line 259-260: Why does a positive change denote a heating effect? You also have to clarify the heating effect on what.

We have made clarifications in the text as follows.

Line 265 - Figure 3 shows the impact of the perturbations on the radiative flux at the top of the atmosphere (perturbation experiment minus the reference case), where a positive change denotes *an increase in the Earth's energy imbalance (a generalized heating effect)*, and a negative change denotes *a decrease in the Earth's energy imbalance (a generalized* cooling effect).

Line 272: (2019-1750) here is ambiguous.

Changed to improve clarity.

Figure 3: It is very difficult to understand the meaning of labels of y-axis.

Thank you for the comment. We have removed the y-axis labels since we already have titles defining each subplot. Also, Table 3 (now referenced in the Figure 3 caption) defines each radiative flux term.

Section 3.3: I suggest authors reorganize this section, because there are some repeated discussions, such as the impacts of emission height on SO2 and SO4.

We have reorganized this section to keep discussion on SO4 and SO2 flow more smoothly. Essentially, we have moved the discussion on boundary layer height to the end of this section since it is distinct from the rest of the discussion.

Line 304-305: Can you explain why does SO2 emitted at height result in decreased SO2 deposition and an increase in oxidation to sulfate?

We have reworded the text as follows to better explain this phenomenon.

Line 314 - In summary, we find that *as SO2 is emitted at height, dry SO2 deposition decreases as the overall lifetime of SO2 in the atmosphere increases (Figure 1d). The longer atmospheric residence time, in turn, increases chemical conversion of SO2 to SO4, which subsequently causes an increase in SO4 in the atmosphere (Figure 6b).*